# Takotsubo cardiomyopathy in patients suffering from acute non-traumatic subarachnoid hemorrhage—A single center follow-up study

Csilla Molnár[1], Judit Gál[1], Dorottya Szántó[1], László Fülöp[2], Andrea Szegedi[2], Péter Siró[1], Endre V. Nagy[3], Szabolcs Lengyel[4], János Kappelmayer[5], Béla Fülesdi[1] *

1 Department of Anesthesiology and Intensive Care, Faculty of Medicine, University of Debrecen, Debrecen, Hungary, 2 Department of Cardiology, Faculty of Medicine, University of Debrecen, Debrecen, Hungary, 3 Department of Internal Medicine, Faculty of Medicine, University of Debrecen, Debrecen, Hungary, 4 Centre for Ecological Research, Hungarian Academy of Sciences, Budapest, Hungary, 5 Department of Laboratory Medicine, Faculty of Medicine, University of Debrecen, Debrecen, Hungary

* fulesdi@med.unideb.hu

**Data Availability Statement:** All relevant data are within the paper and its Supporting information files.

## Abstract

### Background

Takotsubo cardiomyopathy (TTC) is an important complication of subarachnoid hemorrhage (SAH), that may delay surgical or endovascular treatment and may influence patient outcome. This prospective follow-up study intended to collect data on the prevalence, severity, influencing factors and long-term outcome of TTC in patients suffering from non-traumatic SAH.

### Methods

Consecutive patients admitted with the diagnosis of non-traumatic SAH were included. Initial assessment consisted of cranial CT, Hunt-Hess, Fisher and WFNS scoring, 12-lead ECG, transthoracic echocardiography (TTE), transcranial duplex sonography and collecting laboratory parameters (CK, CK-MB, cardiac troponin T, NT-proBNP and urine metanephrine and normetanephrine). Diagnosis of TTC was based on modified Mayo criteria. TTC patients were dichotomized to mild and severe forms. Follow-up of TTE, Glasgow Outcome Scale assessment, Barthel's and Karnofsky scoring occurred on days 30 and 180.

### Results

One hundred thirty six patients were included. The incidence of TTC in the entire cohort was 28.7%; of them, 20.6% and 8.1% were mild and severe, respectively. TTC was more frequent in females (30/39; 77%) than in males (9/39; 23%) and was more severe. The occurrence of TTC was related to mFisher scores and WFNS scores. Although the severity of TTC was related to mFisher score, Hunt-Hess score, WFNS score and GCS, multivariate analysis showed the strongest relationship with mFisher scores. Ejection fraction differences between groups were present on day 30, but disappeared by day 180, whereas wall

**Funding:** Hungarian Brain research program [grant number KTIA_13_NAP-A-II/5], the founder is the Hungarian Academy of Sciences. The funders had no role in study design, data collection and analysis, decision to publish, or preparation of the manuscript.

**Competing interests:** The authors have declared that no competing interests exist.

**Abbreviations:** aSAH, aneurysmal subaracnoid hemorrhage; CT, computed tomography; ECG, electrocardiogram; EF, ejection fraction; GCS, Glasgow Coma Score; ICU, intensive care unit; LV, left ventricle; RV, right ventricle; SAH, subarachnoid hemorrhage; TCCD, Transcranial color-coded duplex; TTC, takotsubo cardiomyopathy; WFNS, World Federation of Neurological Society.

motion score index was still higher in the severe TTC group at day 180. By the end of the follow-up period (180 days), 70 (74.5%) patients survived in the non-TTC, 22 (81.5%) in the mild TTC and 3 (27%) in the severe TTC group (n = 11) (p = 0.002). At day 180, GOS, Barthel, and Karnofsky outcome scores were higher in patients in the control (non-TTC) and the mild TTC groups than in the severe TTC group.

## Conclusions

Takotsubo cardiomyopathy is a frequent finding in patients with SAH, and severe TTC may be present in 8% of SAH cases. The severity of TTC may be an independent predictor of mortality and outcome at 6 months after disease onset. Therefore, a regular follow-up of ECG and TTE abnormalities is warranted in patients with subrachnoid hemorrhage for early detection of TTC.

## Trial registration

The study was registered at the Clinical Trials Register under the registration number of NCT02659878 (date of registration: January 21, 2016).

## Introduction

First described in a case series of Japanese patients in 1991, Takotsubo cardiomyopathy (TTC) represents a reversible left ventricular dysfunction, that mimicks acute coronary syndrome, without the evidence of obstructive coronary artery disease [1]. In the past 3 decades several synonimes were used for TTC, including "apical balloning", "stress-induced cardiomyopathy", "broken heart syndrome" and "transient ventricular ballooning syndrome" [2]. Its estimated incidence is 100 new cases per 1 million inhabitants per year [3] and the prevalence of the syndrome ranges from 1.7 to 2.2% among patients admitted with suspected coronary artery syndrome [4]. The most important causative factors are intense emotional events and any form of physical stress, including severe medical illness. Neurological emergencies represent a large group of the underlying diseases; among them most frequent are subarachnoid hemorrhage, ischemic stroke and epileptic status [5–7]. Subarachnoid hemorrhage (SAH) is the most common neurological cause, making up one-third of the cases. [8]. The prevalence of TTC in SAH ranges from 1.2% up to 26% in different studies [9–12]. The relatively wide range of the prevalence might be in part explained by the changing definition of TTC over the past decades. Recent evidence suggest that prospective studies using predetermined time intervals for echocardiography and taking apical and mild cases also into account may be necessary to evalute the real incidence [13–15]. The importance of TTC in SAH is underlined by the fact that it may lead to neurogenic pulmonary edema, necessitate hemodynamic and ventilatory support and thus has the potential to hamper the treatment especially in severe subarachnoid hemorrhage cases with cerebral vasospasm or raised intracranial pressure. Additionally, TTC may delay surgical or endovascular treatment. Up till now, the majority of the knowledge about the natural history of TTC among patients with SAH has been gathered from retrospective analysis of databases. Prospective long-term follow-up studies on the prevalence and clinical outcome of TTC are scarce.

In the present study we collected data prospectively on the prevalence, severity, influencing factors and long-term outcome of TTC in patients suffering from non-traumatic subarachnoid hemorrhage.

## Subjects and methods

This prospective follow up study was performed at the 8-bed neurosurgical ICU of the Department of Anesthesiology and Intensive Care, University of Debrecen. The ICU serves as a regional center for managing acute subarachnoid hemorrhages being responsible for approximately 2 million inhabitants. During the period between March 2017 and December 2018, all patients who were admitted for SAH were checked for eligibility. The study was registered at the Clinical Trials Register under the registration number of NCT02659878 (date of registration: January 21, 2016). The protocol was approved by the Institutional Ethics Committee of the University of Debrecen (registration number: DE RKEB/IKEB:4317–2015). All included patients or their closest relatives gave written informed consent before participation.

Inclusion criteria were acute subarachnoid hemorrhage (SAH) in adult (age >18 years) patients, provided that the patient arrived at the hospital not later than 48 hours after symptom onset. Patients with head trauma, angioma or A-V malformations were excluded. Patients with known myocardial diseases, including myocardial infarction, heart failure, known structural heart diseases (severe, clinically significant valve insufficiency, and /or significant stenosis) preexisting myocarditis, known coronary artery stenosis necessitating ballon angioplasty, hypertrophic cardiomyopathy and phaechromocytoma were excluded.

Decision on the treatment modality in the acute phase (coling or surgical clipping) was based on consultation with the neurosurgeon and neurointerventionist. Non-surgical and non-interventional therapy at the ICU was performed according to the local protocol that includes maintenance of normovolemia, normocapnia, normoxemia, normomagnesemia and normoglycemia. The use of nimodipine (6x60 mg per os) and analgosedation was based on international guidelines [16]. During the course of the disease, surgical interventions or placing intraventricular or lumbar drains were performed as required.

The diagnosis of TTC was based on the modified Mayo criteria [17], taking the following aspects into account:

- Transient LV/RV wall motion abnormalities, usually extending beyond a single epicardial coronary artery distribution

- Evidence of myocardial involvement by biomarker elevation: cardiac troponin I, creatine kinase, brain natriuretic peptide; N-terminal prohormone of brain natriuretic peptide

- ECG abnormalities: ST elevation/depression, negative T-waves, new bundle brunch blocks

- Potential coronary artery culprit ruled out

Diagnosis of TTC was first verified or excluded in each SAH patient by performing the investigations listed above within 24 hours after admission.

On admission and on the subsequent 6 days (days 1 trough 7) the following investigations were performed on a daily basis: 12-lead ECG; cardiac biomarkers (cardiac troponin T, creatine kinase, N-terminal prohormone of brain natriuretic peptide), transcranial color-coded duplex sonography with blood flow velocity measurements, Glasgow Coma Score (GCS). World Federation of Neurological Society (WFNS) scores, Fisher and Hunt-Hess grading were performed at admission. Collection of urine for determination of urine metanephrine and normetanephrine levels, and transthoracic echocardiography was performed on day 1 and

day 7. If new ECG abnormalities occured between days 1 and 7, additional echcardiography was performed immediately.

During the *30 day follow up* visit the following diagnostic measures were taken: transthoracic echocardiography; 12-lead ECG; New York Heart Association score (NYHA score) assessment; Glasgow Outcome Scale assessment; Barthel's and Karnofsky indices assessments. Additionally, patients were asked to perform a 24-hours timed collection of urine for determination of urine metanephrine and normetanephrine levels one day prior to coming for checkup. If cardiac wall motion abnormality was present on echocardiography, cardiac stress test, CT coronary angiography or conventional coronary angiography were performed as required.

During *180 day follow up visit* the same tests and examinations were performed as on day 30.

Transthoracic echocardiography was performed by two experienced cardiologists (LF, ASz) by Mindray TE7 device (P4-2s 3.5 MHz harmonic imaging transducer) at the initial hospitalization and during follow-up. Investigators were unaware of patient's other parameters (laboratory and ECG results referring to TTC), they only knew that the patient suffered from SAH. During echocardiographic evaluation in 2D-mode the parasternal short axis and apical two- and three-chamber views have been visualized to evaluate left ventricular systolic and end diastolic diameters and wall motion abnormalities. Ejection fraction was calculated by Simpson's equation.

According to the international guidelines the diagnosis of TTC was based on transthoracic echocardiography. This is merely based on two factors: hypo- or akinesis of at least one or more segments of the heart and decreased ejection fraction. The wall motion abnormalities were defined by using the 16-segment model. Segmental wall motion abnormalities have been scored according to the usual method: 1- normokinesis, 2- hypokinesis, 3- akinesis, 4 -dyskinesis, 5 -aneurysm. In order to describe the number and the severity of the affected myocardial regions, the wall motion score index (WMSI) was calculated by dividing the total of the wall motion scores of each segment by 16. [18]. We also took the ejection fraction into account for the differentiation between mild (EF$\geq$40%) and severe TTC (EF <40%). Ejection fraction and wall motion abnormalities have also been detected during the follow-up (30 and 180 days after the initial hospitalization).

Transcranial color-coded duplex (TCCD) sonography was performed by two experienced investigators (BF, PS) using the 2 MHz sector transducer of the GE Venue Go (GE Healthcare 9900 Innovation Drive Wauwatosa, WI 53226 U.S.A.) ultrasound device. He was unaware of the laboratory parameters and the other results, but knew that the patient suffered from SAH. The transtemporal window was insonated and orientation within the structure of the circle of Willis occurred in color mode. After identifying the different vessels of the circle, daily measurements of the middle cerebral, anterior cerebral and posterior cerebral arteries were performed on both sides. Regular TCCD measurements were registered in all patients between days 1 through 7. If ultrasound signs of vasospasm were present in any of the cases, duplex sonographies were performed until day 21 on a daily basis. Cerebral blood flow velocities, pulsatility indices and S/D ratios (systolic velocity/diastolic velocity) were documented in all cases. Based on previous suggestions, vasospasm was considered if mean blood flow velocity was higher than 120 cm/s and severe vasospasm was diagnosed if mean blood flow velocity exceeded 200 cm/s [19].

## Laboratory tests

Creatine kinase activity was determined by a UV kinetic assay utilizing hexokinase and glucose-6 phosphate dehydrogenase enzymes after the rate limiting CK step. CK-MB was

expressed as CK-MB activity by using immunoinhibition with antibodies to the CK-M subunit and subsequently measuring the CK activity as described above by using Roche reagents.

The high sensitive Troponin T assay was measured by an electrochemiluminescent (ECLIA) immunoassay on a Cobas e411 analyzer.

An ECLIA immunoassay was used measuring the N-terminal portion of the pro B-type natriuretic peptide (NT-pro-BNP) that has a considerably longer half life than the active peptide and thus is more suitable for risk stratification.

Metanephrines are usually present in the urine in the form of glucuronate- or sulphate-conjugates, thus as the first step an acid hydrolysis was done. Subsequently, urine samples were diluted by a neutralizing buffer and the measurement was carried out by an isocratic HPLC using an electrochemical detector (ABL and E-Jasco system).

## Statistical analysis

To analyse r×c contingency tables, we used χ2-tests. When 2×2 tables were analysed, we applied Yates' correction for continuity. When the number of expected cases was too small ($< 5$) in more than 20% of the cells, we used a χ2-test with a simulated p-value calculated in a Monte-Carlo randomisation based on 2000 replicates as implemented in the 'chisq.test' function of R.

For the analysis of continuous variables, we checked the homogeneity of variances with the Bartlett test and checked the normal distribution of variables by the Shapiro-Wilks test. In the analysis of variables that were non-normally distributed and/or had heterogeneous variances, we used non-parametric statistical tests. In such cases, we applied Kruskal-Wallis tests, i.e., one-way non-parametric ANOVAs based on ranks to analyse differences between the study groups.

To analyse the effect of one or more continuous independent variables such as age on binary dependent variables such as the occurrence of TTC (no/yes), we applied binary logistic regression using the 'glm' function of R and specifying binomial error distribution. To analyse the effect of one or more continuous independent variables on ordinal variables (that have qualitative levels that can be ordered), such as the severity of TTC (no TTC, mild and severe TTC), we applied ordinal logistic regression using the 'polr' function of R. When there were more than one independent variables, we first ran the full logistic regression model with all variables, and then removed non-significant ($p < 0.05$) variables in a backward stepwise fashion. We report z values (binomial regression), t values (ordinal regression) and odds ratios for logistic regression tests.

To analyse the effect of study group and continuous independent variables on continuous dependent variables, we applied general linear models (GLM) using the 'lm' function of R. For such tests, we report contrasts with the control group as reference (for study group) and slope coefficients ± standard errors (for continuous independent variables) and F values along with p values.

For all statistical analyses, we used the R statistical environment, version 3.6.3. (R Core Team 2020). For all graphs, we used the 'ggplot2' package of R.

## Results

One hundred thirty six patients were included to the study. The flowchart in Fig 1 shows the study cohort, inclusion and exclusion, as well as the numbers of patients during follow-up. General patient characteristics and confounding factors are summarized in Table 1. The most important characteristics of subarachnoid hemorrhage in the entire cohort is shown in Table 2. In the entire cohort endovascular treatment was performed in 73, surgical clipping in

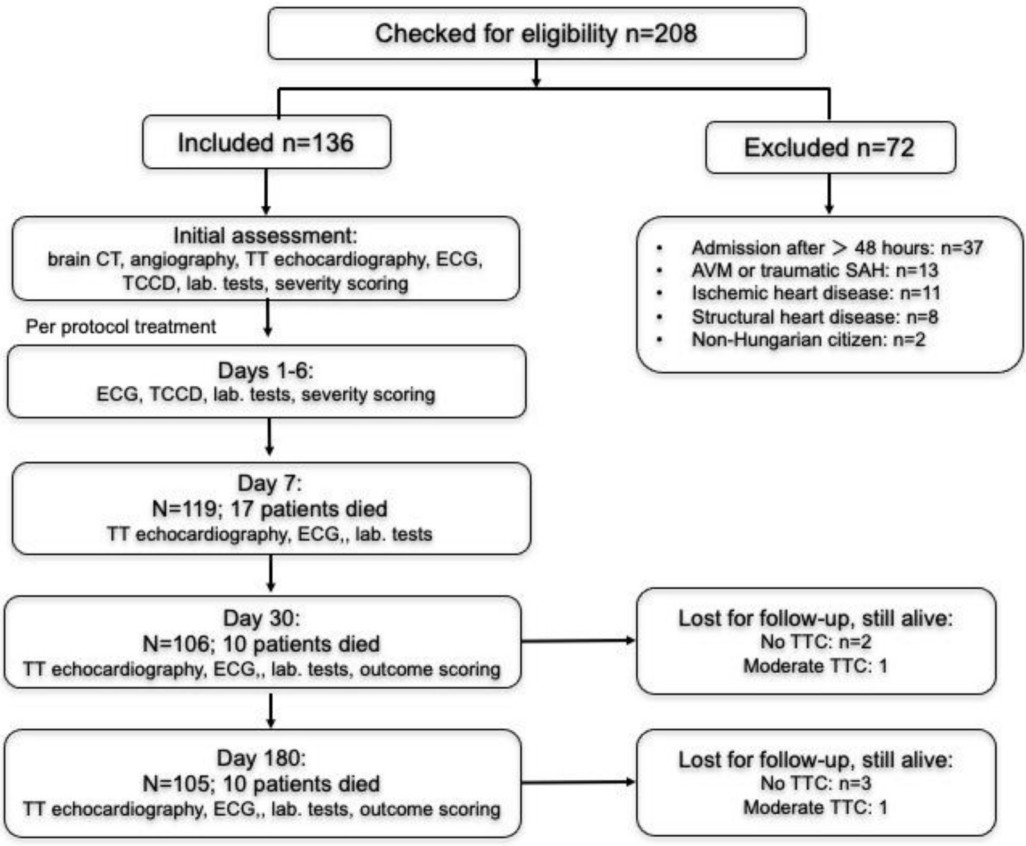

**Fig 1. Flowchart of patient inclusion and exclusion.** CT indicates computed tomography; TT indicates transthoracic; ECG indicates electrocardiogramm; TCCD indicates transcranial color-coded duplex sonography.

21 patients, in 42 patients no intervention was performed. A placement of an intraventricular drain was necessary in 40 cases, whereas lumbar drain was placed in 5 cases.

## The incidence of TTC in the studied cohort

Takotsubo cardiomyopathy developed in 39 of the 136 patients (28.7%). TTC was mild or severe in 28 (20.6%) and 11 (8.1%) patients, respectively.

**Table 1. General characteristics and confounding factors in the entire cohort.**

| Parameter | Value |
|---|---|
| Age (years) | 76.2 (48–72) |
| Gender (F/M) | 80/56 |
| Hypertension (Yes/No) | 73/63 |
| Arrhytmias or conduction disturbances (Yes/No) | 4/132 |
| Hypercholesterinemia/Trigliceridaemia (Yes/No) | 15/121 |
| Diabetes mellitus (Yes/No) | 5/131 |
| Hypothyreosis (Yes/No) | 3/133 |
| Hyperthyreosis (Yes/No) | 1/132 |
| Smoking (Yes/No) | 66/70 |
| Obesity (Yes/No) | 33/103 |

**Table 2. Characteristics of subarachnoid hemorrhage in the entire cohort.**

| Location of the aneurysm ‡ | |
|---|---|
| Internal carotid | 20 |
| Anterior and middle cerebral | 30 |
| Anterior and posterior communicating | 47 |
| Other | 17 |
| No aneurysm | 22 |
| **Ventricular bleeding (Yes/No)** | 69/83 |
| **Parenchymal bleeding (Yes/No)** | 27/109 |
| **Bleeding extent (side)** | |
| Left | 11 |
| Right | 14 |
| Bilateral | 2 |
| No bleeding | 109 |
| **Bleeding location (lobe)** | |
| Frontal | 10 |
| Frontotemporal | 3 |
| Temporal | 7 |
| Temporoparietal | 7 |
| None | 109 |
| **Modified Fisher score at admission** | |
| 1 | 31 |
| 2 | 19 |
| 3 | 27 |
| 4 | 59 |
| Hunt-Hess grade at admission | |
| 1 | 23 |
| 2 | 37 |
| 3 | 31 |
| 4 | 21 |
| 5 | 24 |
| **WFNS score at admission** | |
| 1 | 62 |
| 2 | 18 |
| 3 | 7 |
| 4 | 26 |
| 5 | 23 |
| **GCS at admission** | |
| 3–4 | 11 |
| 5–6 | 12 |
| 7–8 | 16 |
| 9–12 | 10 |
| 13–14 | 25 |
| 15 | 62 |

The distribution of the different patterns according to the recommendation (2) was as follows: classical pattern: n = 10; mid-ventricular pattern: n = 1; reverse pattern: n = 4; focal type: n = 21; atypical pattern (global): n = 3. TTC developed within 7 days after admission in 4 cases, all were focal type.

### The effect of age and gender on the occurence and severity of TTC

The occurrence of TTC was not related to the age of the patients (binomial logistic regression, odds ratio: 1.015, z = 0.821, p = 0.412). Similarly, the severity of TTC was not related to patient age (ordinal logistic regression, coefficient = 0.013 ± S.E. 0.018, odds ratio: 1.013, t = 0.729, p = 0.466).

Takotsubo cardiomyopathy was more frequent in females (30/39; 77%) than in males (9/39; 23%). TTC occurred in 30 (37.5%) of the 80 female patients and in 9 (16%) out of the 56 male patients. The difference was statistically significant (Yates-corrected χ2 = 6.385, df = 1, p = 0.012. TTC appeared to be more severe in female patients: 23 (29%) presented with mild and 7 (9%) with severe TTC, compared to male patients: 5 (9%) mild and 4 (7%) severe TTC cases, respectively (χ2 = 8.152, simulated p = 0.015).

### The effect of comorbidities on the occurrence of TTC

The occurrence of TTC was not influenced by any of the comorbidities studied (S1 Table). A binomial logistic regression calculated in a general linear model testing the effect of each additional illness on the occurrence of TTC (yes/no) found that none of the eight illnesses/conditions influenced the occurrence of TTC.

### The relationship between comorbidities and severity of TTC

The severity of TTC was not influenced by any of the comorbidities studied (S2 Table). An ordinal logistic regression calculated in a general linear model testing the effect of each additional illness on the severity of TTC (none, mild, severe) found that none of the eight illnesses influenced the severity of TTC.

### The relationship between the occurrence of TTC and the severity scores at admission

The incidence of TTC was related to the modified Fisher scores; it increased from 13% in patients with Fisher score of 1 to 16% in score 2, 30% in score 3; 41% in patients with score 4 (Table 3). The occurrence of TTC also increased with WFNS score from 18% in patients with score 1 and to 39% in score 2, 14% in score 3 patients, 35% in score 4 and 48% in score 5 patients (Table 3). None of the other variables influenced the occurrence of TTC (Table 3).

### Relationship between severity of TTC and severity scores at admission

The severity of TTC was related to modified Fisher score, Hunt-Hess score, WFNS score and GCS (Table 4). The incidence of mild TTC was 13%, 11%, 26% and 25% in patients with modified Fisher score of 1, 2, 3 and 4, respectively, whereas the incidence of severe TTC increased from 0 in patients with score 1 to 5% in score 2, 4% in score 3 and 15% in score 4 patients (Table 4).

The incidence of mild TTC varied between 13% (Hunt-Hess score 5) and 29% (H-H score 3) with no obvious trend, whereas the incidence of severe TTC increased from 0 in patients with scores 1 and 2 to 10% in score 3, 5% in score 4 and 29% in score 5 patients (Table 4). The incidence of mild TTC also varied between 17% and 28% in the five WFNS score groups, while the incidence of severe TTC increased from 0 in scores 1 and 3 to 11% in score 2, 8% in score 4 and 30% in score 5 (Table 4). Finally, the incidence of severe TTC was highest (30%) in GCS scores 3 to 6, was lower in scores 7 to 12 and 13 to 14 (8% each) was 0 in score 15; mild TTC varied in frequency between 17 and 27% with no obvious trend (Table 4).

**Table 3. Occurrence of TTC in SAV variable categories.**

| SAV variable | No TTC | TTC | $\chi^2$ |
|---|---|---|---|
| **Location of aneurysm** ‡ | | | |
| Internal carotid | 12 | 8 | |
| Anterior and middle cerebral | 24 | 6 | 2.984 n.s. |
| Anterior and posterior communicating | 32 | 15 | |
| Other | 12 | 5 | |
| No aneurysm | 17 | 5 | |
| **Ventricular bleeding** (Y/N) | 46/51 | 23/16 | 1.059 n.s. |
| **Parenchymal bleeding** (Y/N) | 21/76 | 6/33 | 0.349 n.s. |
| **Bleeding extent (side)** | | | |
| Left | 8 | 3 | 1.311 n.s. |
| Right | 11 | 3 | |
| Bilateral | 2 | 0 | |
| No bleeding | 76 | 33 | |
| **Bleeding location (lobe)** | | | |
| Frontal | 7 | 3 | 2.060 n.s. |
| Frontotemporal | 3 | 0 | |
| None | 76 | 33 | |
| Temporal | 5 | 2 | |
| Temporoparietal | 6 | 1 | |
| **Hunt-Hess grade** | | | |
| 1 | 18 | 5 | 6.856 n.s. |
| 2 | 31 | 6 | |
| 3 | 19 | 12 | |
| 4 | 15 | 6 | |
| 5 | 14 | 10 | |
| **Modified Fisher score** | | | |
| 1 | 27 | 4 | **9.481** * |
| 2 | 16 | 3 | p = 0.024 |
| 3 | 19 | 8 | |
| 4 | 35 | 24 | |
| **WFNS** | | | |
| 1 | 51 | 11 | **9.823** * |
| 2 | 11 | 7 | p = 0.041 |
| 3 | 6 | 1 | |
| 4 | 17 | 9 | |
| 5 | 12 | 11 | |
| **GCS** | | | |
| 3–4 | 6 | 5 | |
| 5–6 | 6 | 6 | |
| 7–8 | 13 | 3 | 13.509 * |
| 9–12 | 4 | 6 | p<0.05 |
| 13–14 | 17 | 8 | |
| 15 | 51 | 11 | |

Significant variables are highlighted in Bold.

Note:

* p < 0.05; n.s. indicates non-significant difference

‡ neighbouring values or similar categories were pooled to meet $\chi^2$-test assumptions.

**Table 4. Severity of TTC in different SAH variable categories.**

| SAV variable | No TTC (n = 97) | Mild TTC (n = 28) | Severe TTC (n = 11) | $\chi^2$ |
|---|---|---|---|---|
| **Location of aneurysm** ‡ | | | | 3.836 n.s. |
| Internal carotid | 12 | 5 | 3 | |
| Middle and anterior cerebral | 24 | 4 | 2 | |
| Anterior and posterior communicating | 32 | 11 | 4 | |
| Other | 12 | 4 | 1 | |
| No aneurysm | 17 | 4 | 1 | |
| **Ventricular bleeding** (Y/N) | 46/51 | 14/14 | 9/2 | 4.684 n.s. |
| **Parenchymal bleeding (Y/N)** | 21/76 | 4/24 | 2/9 | 0.761 n.s. |
| **Bleeding side** | | | | 1.392 n.s. |
| Left | 8 | 2 | 1 | |
| Right | 11 | 2 | 1 | |
| Bilateral | 2 | 0 | 0 | |
| No bleeding | 76 | 24 | 9 | |
| **Bleeding extent (lobe)** | | | | |
| Frontal | 7 | 2 | 1 | 2.779 n.s. |
| Frontotemporal | 3 | 0 | 0 | |
| None | 76 | 24 | 9 | |
| Temporal | 5 | 1 | 1 | |
| Temporoparietal | 6 | 1 | 0 | |
| **Hunt-Hess grade** | | | | **22.687** ** p = 0.004 |
| 1 | 18 | 5 | 0 | |
| 2 | 31 | 6 | 0 | |
| 3 | 19 | 9 | 3 | |
| 4 | 15 | 5 | 1 | |
| 5 | 14 | 3 | 7 | |
| **Modified Fisher score** | | | | **12.668** * p = 0.047 |
| 1 | 27 | 4 | 0 | |
| 2 | 16 | 2 | 1 | |
| 3 | 19 | 7 | 1 | |
| 4 | 35 | 15 | 9 | |
| **WFNS score** | | | | |
| 1 | 51 | 11 | 0 | **24.258** ** p = 0.003 |
| 2 | 11 | 5 | 2 | |
| 3 | 6 | 1 | 0 | |
| 4 | 17 | 7 | 2 | |
| 5 | 12 | 4 | 7 | |
| **GCS** | | | | **22.616** *** p = 0.001 |
| 3–6 | 12 | 4 | 7 | |
| 7–12 | 17 | 7 | 2 | |
| 13–14 | 17 | 6 | 2 | |
| 15 | 51 | 11 | 0 | |

Significant effects are highlighted in Bold.

Note:

* $p < 0.05$;

** $p < 0.01$;

*** $p < 0.001$

‡ neighbouring values or similar categories were pooled to meet $\chi^2$-test assumptions.

When the effects of all SAV variables were tested simultaneously in a binomial logistic regression followed by backward stepwise removal of non-significant effects, only the modified Fisher score influenced the incidence of TTC, and a positive coefficient indicated that TTC was more likely to occur in patients with higher modified Fisher scores (estimate: 0.54 ± S.E. 0.185, odds ratio: 1.719, z = 2.923, p = 0.003). Similarly, the severity of TTC showed a positive relationship with only the modified Fisher score (estimate: 0.57 ± 0.185, odds ratio: 1.764, z = 3.071, p = 0.036).

## Relationship between development of vasospasm and TTC

Vasospasm (any type) occurred in 23/97 patients in the control (non TTC) group and 13/39 in the TTC group (Chi2:0.74; P = 0.36), indicating that TS was not more frequent in SAH patients in whom vasospasm developed. Severe vasospasm occurred in 5/97 patients and 6/39 patients in the non-TTC and TTC groups, respectively (Chi2:2,87; p = 0.09).

## Laboratory parameters

For CK, the differences between the three groups were significant on day 1 (Kruskal-Wallis $\chi 2 = 11.642$, df = 2, p = 0.003), day 2 ($\chi 2 = 8.673$, df = 2, p = 0.013) and day 4 ($\chi 2 = 7.478$, df = 2, p = 0.024) but not on the other days. For CK-MB, the differences were significant on day 1 ($\chi 2 = 9.346$, df = 2, p = 0.009), but not on the other days. For CTNT, the differences were highly significant each day, mostly because of high values in the severe TTC group (Fig 2).

For NTPBNP, the differences were also highly significant each day because of high values in the severe TTC group (Fig 3).

General linear models showed that the level of metanephrin in the urine did not differ between the three study groups and was not influenced by the level of norepinephrine (arterenol) or dobutamin administered, and this was true for the day of admission, as well as days 30 and 180 of follow up. In contrast, the level of normetanephrin differed between TTC groups and was influenced by norepinephrine (arterenol) and dobutamin administration (Fig 4, S3 Table). The interaction between groups and norepinephrine (arterenol) or dobutamin were not significant, indicating that the effects of the drugs were similar in each group. The difference between study groups was marginally non-significant 30 days after the bleeding, partly due to a slight difference between the control and the mild TTC group and partly due to few data in the severe TTC group (Fig 4B, S3 Table). Finally, there was no difference in

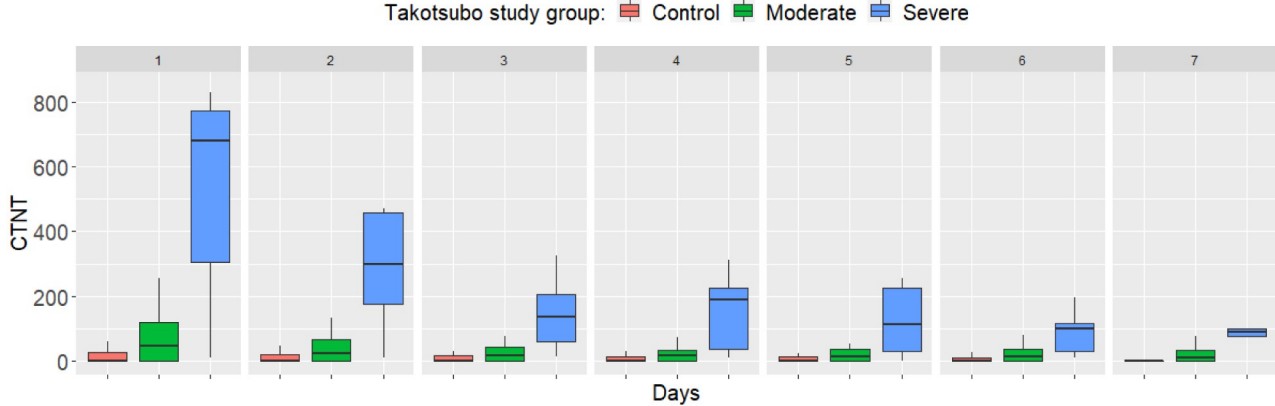

**Fig 2. Differences in cardiac troponin T (CTNT) between study groups from day 1 to 7.** Boxplots show the median, upper and lower quartiles, and minimum and maximum values, with outliers omitted for clarity. Differences are highly significant each day (p<0.001).

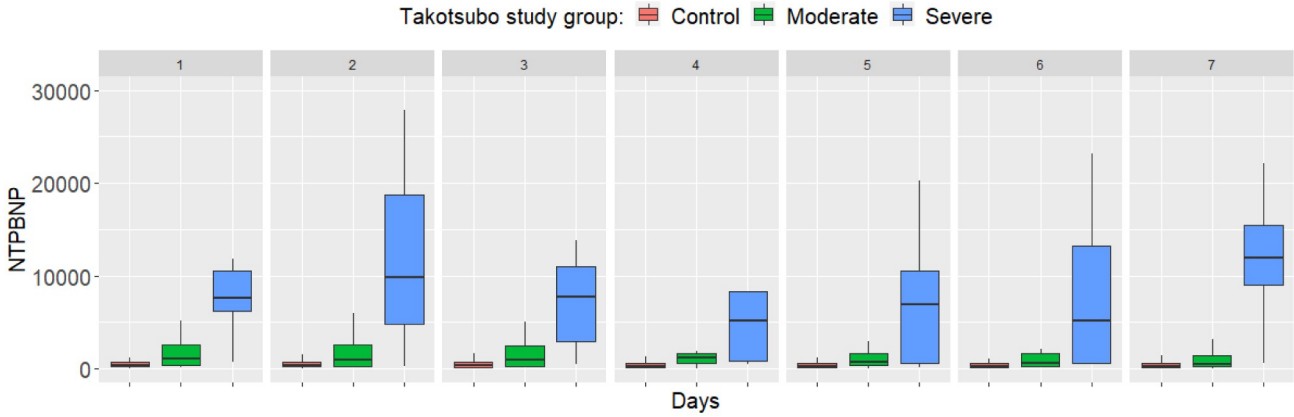

**Fig 3. Differences in N-terminal pro-brain-type natriuretic peptide (NTPBNP) between study groups from day 1 to 7.** Boxplots show the median, upper and lower quartiles, and minimum and maximum values, with outliers omitted for clarity. Differences are highly significant each day (p<0.001).

normetanephrin concentration between the study groups 180 days after the bleeding (GLM, $F_{2,54} = 0.209$, p = 0.812).

## Changing the echocardiography parameters over time

The ejection fraction was significantly lower in patients with severe TTC than in non-TTC controls, and was intermediate but closer to level of controls in patients in patients with mild

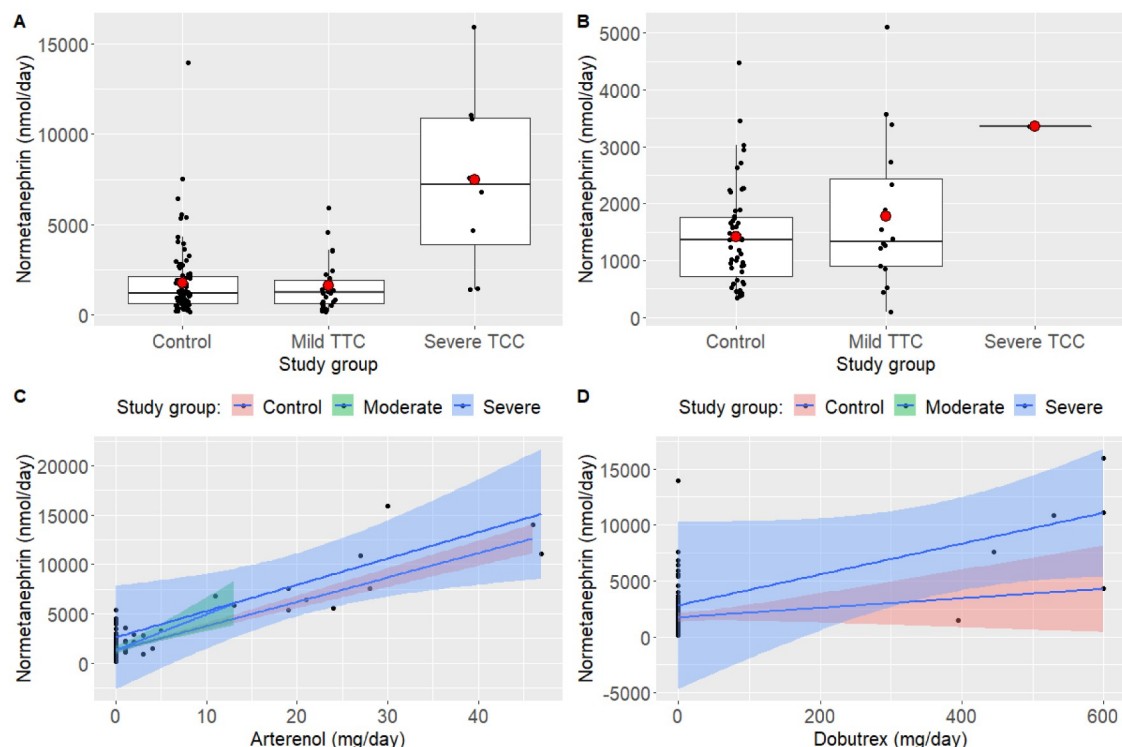

**Fig 4. Urine normetanephrin levels on the day of admission (A) and 30 days later (B), and the effect of norepinephrine (arterenol) (C) and dobutamin (dobutrex) (D) on normetanephrin concentrations on the day of admission.** Boxplots show datapoints (black dots jittered and outliers omitted for clarity), median, upper and lower quartiles, and minimum and maximum values; red dots indicate means. Shaded areas in C and D indicate 95% confidence intervals. Note that the scales are different on the ordinates; on the day of admission (A), median normetanephrine concentration is 7-fold higher in the severe TTC group.

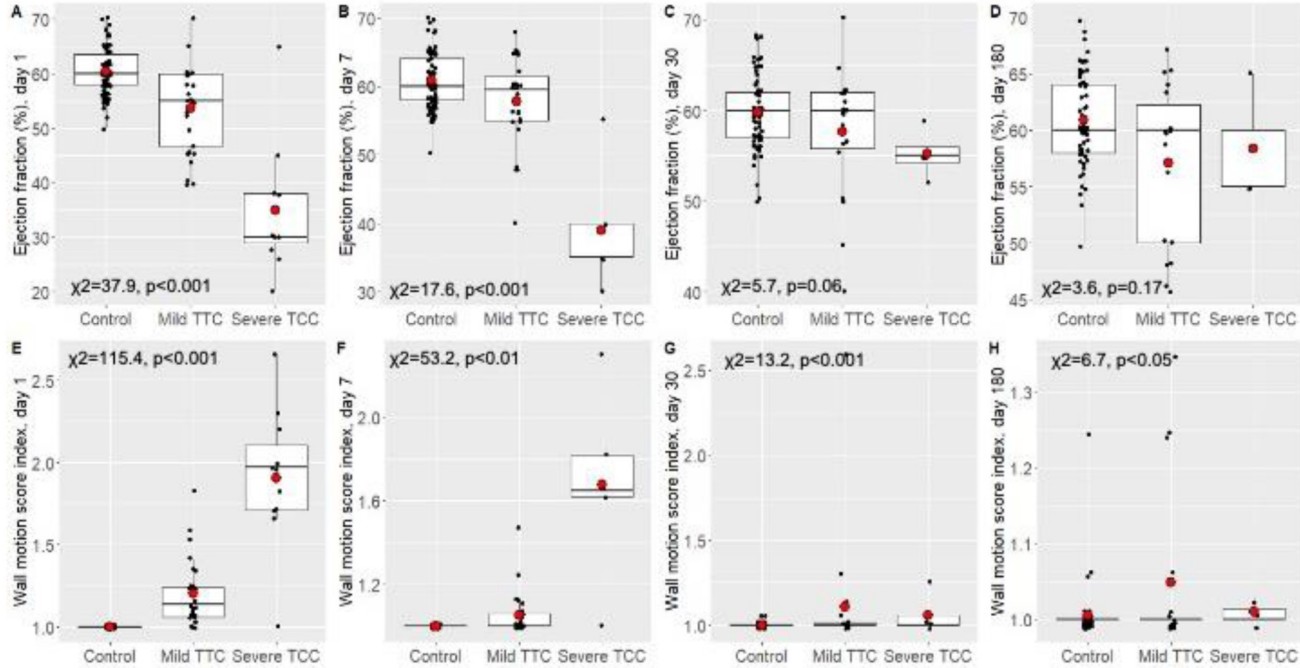

**Fig 5. Differences in ejection fraction (%) (A-D) and wall motion score index (E-H) in the study groups on day 1, day 7, day 30 and day 180.** Boxplots show datapoints (black dots; jittered and outliers omitted for clarity), median, upper and lower quartiles, and minimum and maximum values; red dots indicate means.

on both day 1 and 7 (Fig 5A–5D). These differences were marginally significant on day 30 and disappeared by day 180. Thus, ejection fraction improved stepwise in TTC patients during the follow-up period, but there were still signs of weak contractile function.

The wall motion score index was highest in patients in the severe TTC group, and low in the mild TTC and control groups (Fig 5E–5H). The differences were significant on all study days, indicating that the difference could be detected even after 180 days.

## Relationship between severity of TTC and mortality

By the end of the follow-up period (180 days), 70 (74.5%) patients survived in the control (non-TTC) group (n = 94), 22 (81.5%) survived in the mild TTC group (n = 27) and three (27%) survived in the severe TTC group (n = 11). The difference in mortality rate was significant ($\chi2$ = 12.395, df = 2, p = 0.002). The number of deceased patients during different phases of follow up were as follows:

- Days 1–7: NoTTC group: n = 9; mild TTC: n = 2, severe TTC: n = 6

- Days 8–30: No TTC: n = 6; mild TTC: n = 3; severe TTC: n = 1

- Days 31–180: No TTC: n = 9; mild TTC: n = 0; severe TTC: n = 1

## Relationship between the severity of TTC and the outcome of the patients

GOS, Barthel, and Karnofsky outcome scores were higher in patients in the control (non-TTC) and the mild TTC groups than in the severe TTC group (Fig 6). All the differences were significant, with the exception of the differences in Barthel scores during the second check-up (Fig 6E), indicating that severe TTC is associated with worse long-term clinical outcome of the patients.

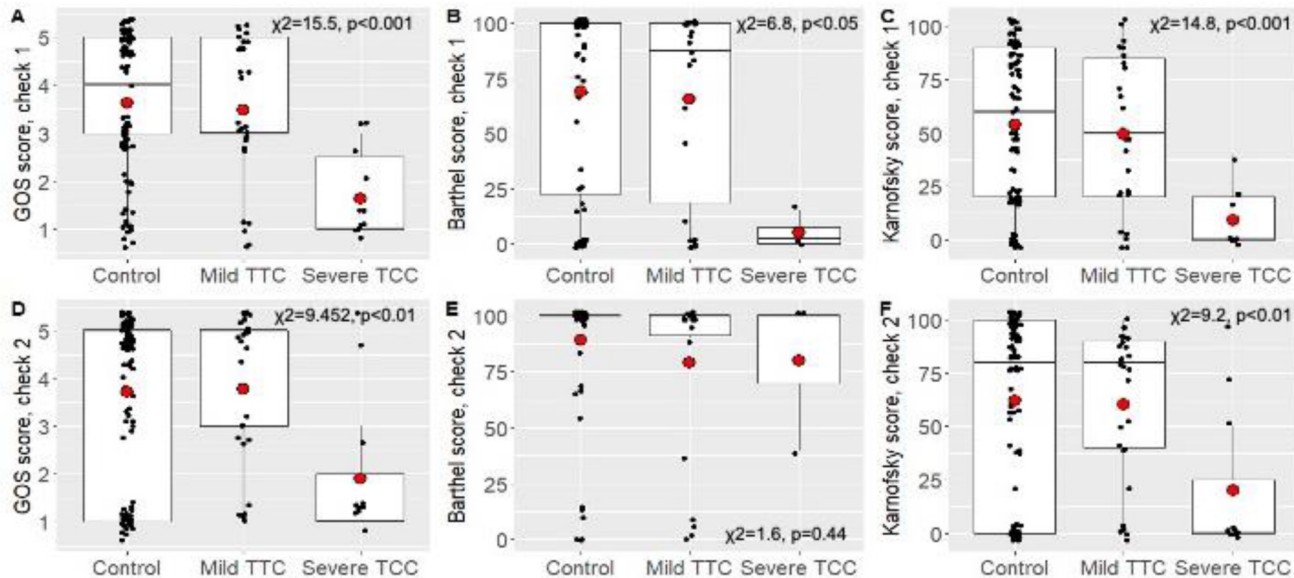

**Fig 6. Differences in GOS, Barthel and Karnofsky scores between study groups at the first check-up at 30 days (A-) and the second check-up at 180 days (D-F).** Boxplots show datapoints (black dots; jittered and outliers omitted for clarity), median, upper and lower quartiles, and minimum and maximum values; red dots indicate means.

## Discussion

The present report is among the first prospective studies assessing the incidence, predisposing factors and long-term clinical outcome of patients with TTC after aneurysmal subarachnoid hemorrhage (aSAH) in a parallel, complex fashion. Our findings contribute to better understanding the natural history of TTC after aSAH and suggests a link between pathophysiological factors, clinical presentation of TTC and outcome parameters. As all aSAH patients were consecutively included, the reported incidence in our cohort reflects the real incidence of TTC according to recent recommendations. An important novelty is the systematic, prospective, long-term (180 days) follow up of the echocardiographic parameters in parallel with the cardiac biomarkers and time-collected urine catecholamine concentrations. Furthermore, based on the predefined ejection fraction threshold of 40%, we were able to define mild and severe forms of TTC in the acute phase of subarachnoid hemorrhage that showed a good relationship with long term outcome of the patients. We were the first demonstrating that cardiac wall motion abnormalities in severe TTC cases may persist for as long as 6 months after SAH onset.

We found that cardiac wall motion disturbances can be detected in 28.7% of the cases. TTC may influence long-term mortality, and severe TTC may affect long term outcome. The prevalence of TTC varied in a wide range in previous reports, reaching up to 30% [5]. The differences could be, at least in part explained by developments in the definition of TTC over time: typical classical form of TTC was found in 2.2% of SAH patients in the study of Talahma [20], while others reported on 8.8% [9], 11% [10], 22% [21] and 27% [11] wall motion abnormalities. The reported incidence may have been influenced by the nature of the study (prospective or retrospective) and the timing and frequency of echocardiographic assessments. The prevalence of TTC in SAH may be underestimated by the majority of the studies [22] and prospective studies using regular echocardiography follow-up have been encouraged [15]. In the present study we included only patients in whom transthoracic echocardiography was performed within 24 hours after admission, and ECG and cardiac biomarkers were regularly

checked on the daily basis. Whenever a significant ECG change was found, echocardiography was performed immediately, in addition to the day 1 and day 7 examinations. Therefore, we believe that the 28.7% incidence rate of TTC in our cohort reflects the true incidence of this condition; all cases of newly appearing wall motion abnormality were included. During the off-line analysis, we arbitrarily defined a mild and a severe group of TTC patients based on the ejection fraction; the predefined threshold of 40% was based on clinical experience that showed that below this level the cardiovascular status of the patients usually worsens and pharmacologic intervention is often necessary. Based on this grouping, we found the incidence of severe TTC 8.1% in SAH, which corresponds to the prevalence rates that have been published previously [9].

In line with previous studies we also found that TTC in our cohort was more frequent in females than in males, and more severe cases were found among women. In two retrospective studies, the percentage of females were 100% [20] and 78% [5], respectively. In a prospective study of Kothavale et al. [22] the represansation of females was 68%. To explain these differences it is proposed that estrogen deficiency in postmenopausal women results in an increased sympathetic drive and endothelial dysfunction, making cardiac microcirculation more vulnerable during critical illnesses [23].

In contrast to previous reports, we could not find any relationship between occurrence of TTC and any of the cardiovascular risk factors and comorbidities. This may be expained by a relatively high prevalence of the cardiovascular risk factors in the Hungarian general population.

We found that the incidence of TTC was associated with modified Fisher scores and WFNS scores. Moreover, the severity of TTC was related to higher modified Fisher scores, Hunt&Hess scores, WFNS scores and GCS. This indicates that the more severe is the SAH at admission, the more severe TTC develops during the course of the disease. Similar observations were published by Talahma et al. [20]. Malik et al. also found a positive association between Hunt&Hess scores and left ventricular ejection fraction [24]. Kothavalet et al. [22] and Kilbourn et al. [25] found that Hunt&Hess grade 3–5 has a predictive value for the presence of regional wall motion abnormalities. It has to be noted, that in the present study binominal regression testing revealed that of the severity scores, only the modified Fisher score on admission was strongly associated with severity of TTC.

Follow-up investigations of cardiac biomarkers showed that both cardiac troponin and NT-proBNP values were elevated from the day of admission both in the mild and the severe TTC groups. While troponin T and NT-proBNP levels tended to normalize in the mild group from day 3 on, in the severe group higher levels of these cardiac biomarkers persisted by day 7. The time pattern of the elevation of cardiac troponin is in line with previous observations [24, 26] confirming its high sensivity to detect left ventricular dysfunction in SAH. Although elevated serum BNP was significantly associated with regional wall motion abnormalities, it was not associated with 30- day mortality in previous studies [27, 28].

It is widely accepted that catecholamines play a key role in the development of TTC. Elevation of plasma catecholamine (epinephrine and norepinephrine) concentrations are typical findings in patients with TTC [29–31]. At rest, the source of norepinephrine (NE) acting on the cardiac receptors is the andrenal medulla in 2–8% percent, and the rest is released by the sympathetic nerve endings [32]. In the acute phase of TTS, an increased concentration of circulating catecholamines, and an increased production of catecholamines at the myocardial level has been observed. Further, the density of the beta-2 receptors is higher in the apical than in the basal part of the ventricle, whereas beta1-adrenergic receptors are expressed predominantly at the base. A physiologic rise in the levels of epinephrine results in a switch of beta2-adrenoceptor coupling from Gs to Gi, in order to prevent the proapoptotic effect of the intense

stimulation of the beta-1 adrenoceptors. Additionally, switching of the coupling mechanism leads to a negative inotropic effect on the apex of the left ventricle [33, 34]. Additionally, increased cardiac sympathetic activity may cause endothelial dysfunction and consequent coronary microvascular constriction, thus contributing to myocardial ischemia [23]. In accordance with these, in the present study we found that metanephrine concentration in the urine did not differ between TTC and non-TTC patients, whereas normetanephrine was elevated in patients with TTC at admission. This difference was missing at day 30 and day 180 of the follow up. Normetanephrine concentrations were not influenced by dobutamine or norepinephrine treatment. In line with these observations, Akashi et al. demonstrated an improvement of cardiac sympathetic hyperactivity in TCC patients using myocardial scintigraphy [35]. It is tempting to hypothetise that severe TTC itself, via the systemic effects of myocardial dysfunction, may contribute to the elevation of normetanephrin production and worsening of TTC as part of a vicious circle. However, our data are insufficient to confirm this.

Per definition, wall motion abnormalities are temporary in patients with TTC and show and improvement over time along with the ejection fraction of the heart. Kim and co-workers demonstrated an improvement of the ejection fraction from 38%, to 61% during a 6 week follow-up [36]. In a study of Templin et al. reduced left ventricular ejection fraction (mean value: 40.7) was found in 86.5% of patients with TTC, which showed a recovery over time during a 60-day follow-up [37]. Similarly, Dias et el. reported on an improvement of the left ventricular ejection fraction from the intial 32.1% to 54% in a 6-month follow up study. [38]. The EF tended to normalize in mild cases, while in the severe TTC group low EF persisted at 7 days. Although at 30 days statistically significant differences between the TTC groups compared to non-TTC groups could not be demonstrated, it has to be noted that EF was still lower in severe cases and the lack of statistical difference might be explained by the relatively small sample size. Importantly, wall motion score index differences persisted over 180 days in the severe TTC group, indicating that the contractile function of the heart may not entirely recover at 6 months, despite normalization of the global left ventricular ejection fraction.

The severity of TTC in our cohort significantly affected the outcome and mortality of the patients during the 180 days follow-up. The mortality at 6 months was significantly higher among patients with severe TTC whereas the mortality in mild TTC cases was comparable with that of non-TTC subarachnoid hemorrhage patients. The mortality of the present study (33.3%) is lower than that in the study of Kilbourn (47%) [25], buth slightly higher that reported by Talahma (28%) [20] and Abd (26%) [39]. In those who were alive at the day 180 follow-up, both Glasgow outcome scores and Karnovsky scores were significantly lower in the severe compared to the mild TTC group. Barthel's index reflected worse outcome in the severe TTC group only at 30 days in the present study. There are only few studies that have assessed long-term outcome of TTC. Similarly to our results, Crago et al. demonstrated differences in Barthel's index at 30 days but not at 60 days in TCC patients after subarachnoid hemorrhage [40]. Mutoh et al. reported on poor 3-month functional outcome as assessed by modified Rankin Scale in TTC patients with an EF <40% [41].

We have to mention some limitations of our study. This was a single center follow-up study with an inclusion period of 22 months and as a consequence, the number of included patients is limited. We may also be criticized for the follow-up of vasospasm with transcranial color-coded duplex sonography. In fact, the vasospasm may be accurately diagnosed by angiography, but we decided to perform a regular TCCD check of the blood flow velocities. If clinical and/or ultrasonographic signs of vasospasm were found, cerebral angiography was performed. We did not measure the blood levels of epinephrine and norepinephrine because we belive that sampling blood for these parameters would represent a rapidly changing snapshot, while urine metanephrine and normetanephrine in 24 hour collected urine may be more representative of the

tissue exposure to these hormones. Finally, we arbitrarily defined mild and severe forms of TTC during analysis of the results based on ejection fraction. The rationale behind this dichotomization was that previous reports indicated that the critical threshold of ejection fraction that determines cardiovascular complications and long-term outcome lies between 40–45% [37].

## Conclusion

We found that Takotsubo cardiomyopathy is a frequent phenomenon occurring in patients with aneurysmal subrachnoid hemorrhage, and severe TTC is present in 8% of SAH cases. The incidence of SAH is associated with the modified Fisher and WFNS scores at admission. The severity of TTC may may be an independent predictor of mortality and outcome at 6 months after disease onset. Systematic and regular follow-up of ECG and echocardiographic abnormalities is warranted in patients with subrachnoid hemorrhage for early detection of TTC. Further studies are required to identify the best treatment approach to the cardiac changes which may accompany SAH.

## Supporting information

**S1 Table. Occurrence of TTC by comorbidities of the patients.**
(DOCX)

**S2 Table. Severity of TTC by comorbidities of the patients.**
(DOCX)

**S3 Table. Results of general linear models testing the differences in normetenephrin concentration between study groups and the effect of arterenol and dobutrex on normetanephrin concentration on days 1 and 30.**
(DOCX)

## Author Contributions

**Conceptualization:** Csilla Molnár, László Fülöp, Endre V. Nagy, János Kappelmayer, Béla Fülesdi.

**Data curation:** Judit Gál, Dorottya Szántó, Szabolcs Lengyel.

**Formal analysis:** Szabolcs Lengyel.

**Funding acquisition:** Csilla Molnár.

**Investigation:** Csilla Molnár, Judit Gál, Dorottya Szántó, László Fülöp, Andrea Szegedi, Péter Siró, János Kappelmayer.

**Methodology:** Csilla Molnár, Judit Gál, Dorottya Szántó, Béla Fülesdi.

**Project administration:** Csilla Molnár, Dorottya Szántó.

**Software:** Szabolcs Lengyel.

**Validation:** Judit Gál, Dorottya Szántó.

**Writing – original draft:** Csilla Molnár, Dorottya Szántó, Endre V. Nagy, Szabolcs Lengyel, János Kappelmayer, Béla Fülesdi.

**Writing – review & editing:** Béla Fülesdi.

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
