## [Decision Letter · Decision Letter 0]

8 Mar 2022

PONE-D-21-36989Takotsubo Cardiomyopathy in Patients Suffering From Acute Non-traumatic Subarachnoid Hemorrhage - a Single Center Follow-up StudyPLOS ONE

Dear Dr. Fülesdi,

Thank you for submitting your manuscript to PLOS ONE. First of all, I deeply apologize for this long reviewing process. I was waiting for a delayed review (reviewer 2) but decided to move forward.

Your paper was reviewed by an expert in the field and myself. As shown in the reviewer's comments, your study was well performed, and paper was well written. However, numerous numbers of papers were already published on this topic. I am aware of this journal's policy (i.e., novelty and clinical significance should not be judged) but please try to add some unique features in your study. 

We look forward to receiving your revised manuscript.

Kind regards,

Tomohiko Ai, M.D., Ph.D.

Academic Editor

PLOS ONE

Journal Requirements:

Hungarian Brain research program [grant number KTIA_13_NAP-A-II/5], the founder is the Hungarian Academy of Sciences

Reviewers' comments:

Reviewer's Responses to Questions

**Comments to the Author**

1. Is the manuscript technically sound, and do the data support the conclusions?

Reviewer #1: Yes

2. Has the statistical analysis been performed appropriately and rigorously? 

Reviewer #1: Yes

3. Have the authors made all data underlying the findings in their manuscript fully available?

Reviewer #1: Yes

4. Is the manuscript presented in an intelligible fashion and written in standard English?

Reviewer #1: Yes

5. Review Comments to the Author

Reviewer #1: This single center, prospective study was well conducted, and its results (frequency, identification of risk factors, measurement of cardiac biomarkers, outcome predictability etc.) were consistent with those of previous studies. It should be noted, however, that there have been numerous studies on the clinical characteristics of SAH-induced Takotsubo cardiomyopathy, and this study is just one of them. Few novel findings were derived from this study, except that long-term echocardiographic follow-up had been conducted.

6. PLOS authors have the option to publish the peer review history of their article (what does this mean?). If published, this will include your full peer review and any attached files.

Reviewer #1: No

---

## [Author Response · Author response to Decision Letter 0]

29 Apr 2022

Comments from the editorial office

• Funding: Hungarian Brain research program [grant number KTIA_13_NAP-A-II/5], the founder is the Hungarian Academy of Sciences. The funders had no role in study design, data collection and analysis, decision to publish, or preparation of the manuscript.

• Data availability: In the declarations section we stated: „all available material (including raw data and statistical analysis will be made available upon request from the corresponding author.”

• phrase “data not shown”: we delated this part from the text as suggested.

• Supporting Information files: we included them at the end of the manuscript, as suggested.

Reviewer and editor’s comment 

• Novelty of the study as compared to previous reports. Thank you for this comment. For sake of clarity we added a section to the discussion section that is intended to point out the novelties of the present study. This part of the discussion sections reads now as follows: „The present report is among the first prospective studies assessing the incidence, predisposing factors and long-term clinical outcome of patients with TTC after aneurysmal subarachnoid hemorrhage (aSAH) in a parallel, complex fashion. Our findings contribute to better understanding the natural history of TTC after aSAH and suggests a link between pathophysiological factors, clinical presentation of TTC and outcome parameters. As all aSAH patients were consecutively included, the reported incidence in our cohort reflects the real incidence of TTC according to recent recommendations. An important novelty is the systematic, prospective, long-term (180 days) follow up of the echocardiographic parameters in parallel with the cardiac biomarkers and time-collected urine catecholamine concentrations. Furthermore, based on the predefined ejection fraction threshold of 40%, we were able to define mild and severe forms of TTC in the acute phase of subarachnoid hemorrhage that showed a good relationship with long term outcome of the patients. We were the first demonstrating that cardiac wall motion abnormalities in severe TTC cases may persist for as long as 6 months after SAH onset.”

Again, thank you very much for the thorough review process and for your suggestions. We hope that in its present form the manuscript will meet the requirements of the journal.

---

## [Editor Report · Decision Letter 1]

4 May 2022

Takotsubo Cardiomyopathy in Patients Suffering From Acute Non-traumatic Subarachnoid Hemorrhage - a Single Center Follow-up Study

PONE-D-21-36989R1

Dear Dr. Fülesdi,

We’re pleased to inform you that your manuscript has been judged scientifically suitable for publication and will be formally accepted for publication once it meets all outstanding technical requirements.

Kind regards,

Tomohiko Ai, M.D., Ph.D.

Academic Editor

PLOS ONE
---

## [Editor Report · Acceptance letter]

16 May 2022

PONE-D-21-36989R1 

Takotsubo Cardiomyopathy in Patients Suffering From Acute Non-traumatic Subarachnoid Hemorrhage - a Single Center Follow-up Study 

Dear Dr. Fülesdi:

I'm pleased to inform you that your manuscript has been deemed suitable for publication in PLOS ONE. Congratulations! Your manuscript is now with our production department. 

Kind regards, 

on behalf of

Dr. Tomohiko Ai 

Academic Editor

PLOS ONE